# Urban Treetop Detection and Tree-Height Estimation from Unmanned-Aerial-Vehicle Images

**Hui Wu** [1,†]**, Minghao Zhuang** [1,†]**, Yuanchi Chen** [2]**, Chen Meng** [3]**, Caiyan Wu** [1,4,*]**, Linke Ouyang** [5]**, Yuhan Liu** [3]**, Yi Shu** [6]**, Yuzhong Tao** [6]**, Tong Qiu** [7] **and Junxiang Li** [1]

1   School of Design, Shanghai Jiao Tong University, Shanghai 200240, China; junxiangli@sjtu.edu.cn (J.L.)
2   School of Communication & Information Engineering, Shanghai University, Shanghai 200444, China
3   School of Ecological and Environmental Science, East China Normal University, Shanghai 200241, China
4   Department of Geography, Humboldt-Universität zu Berlin, Rudower Chaussee 16, 12489 Berlin, Germany
5   Shanghai Artificial Intelligence Laboratory, Shanghai 200232, China
6   Department of Energy and Environmental Protection, Bao Iron & Steel Co., Ltd., Shanghai 201900, China
7   Department of Ecosystem Science and Management, Pennsylvania State University, University Park, PA 16802, USA
*   Correspondence: caiyanwu@sjtu.edu.cn
†   These authors contributed equally to this work.

**Abstract:** Individual tree detection for urban forests in subtropical environments remains a great challenge due to the various types of forest structures, high canopy closures, and the mixture of evergreen and deciduous broadleaved trees. Existing treetop detection methods based on the canopy-height model (CHM) from UAV images cannot resolve commission errors in heterogeneous urban forests with multiple trunks or strong lateral branches. In this study, we improved the traditional local-maximum (LM) algorithm using a dual Gaussian filter, variable window size, and local normalized correlation coefficient (NCC). Specifically, we adapted a crown model of maximum/minimum tree-crown radii and an angle strategy to detect treetops. We then removed and merged the pending tree vertices. Our results showed that our improved LM algorithm had an average user accuracy (UA) of 87.3% (SD± 4.6), an average producer accuracy (PA) of 82.8% (SD± 4.1), and an overall accuracy of 93.3% (SD± 3.9) for sample plots with canopy closures less than 0.5. As for the sample plots with canopy closures from 0.5 to 1, the accuracies were 78.6% (SD± 31.5), 73.8% (SD± 10.3), and 68.1% (SD± 12.7), respectively. The tree-height estimation accuracy reached more than 0.96, with an average RMSE of 0.61 m. Our results show that the UAV-image-derived CHM can be used to accurately detect individual trees in mixed forests in subtropical cities like Shanghai, China, to provide vital tree-structure parameters for precise and sustainable forest management.

**Keywords:** unmanned aerial vehicle (UAV); local-maximum algorithm; urban forest; treetop detection; subtropical evergreen–deciduous broadleaved mixed forest





## 1. Introduction

Urban trees refer to the woody perennial plants that thrive in and around cities [1]. Urban trees, as critical components of urban forests, provide diverse ecosystem services, such as air purification [2], habitat [3], food [4], climate regulation [5], cultural value [6], etc. Modern urban forestry requires accurate information on individual trees to maximize the ecological benefits of urban forests. The individual-scale information includes the tree structure, location and spatial distribution, species diversity, and function [7]. Due to rapid urbanization, urban trees exhibit high dynamics in numbers and spatial distributions from year to year, highlighting the need to conduct regular annual forestry inventories for forest management [8]. The accurate biophysical parameters, such as the tree height, canopy width, and tree species, are the key pieces of information for urban tree planning and management [9–12].

However, the traditional urban forest inventory is time-consuming and labor-intensive, and therefore it cannot meet the ever-increasing demands for timely and accurate information on urban trees in current precision forest management [13]. High-spatial- and -temporal-resolution remotely-sensed images acquired via satellite and/or unmanned aerial vehicles (UAVs) can provide accurate and low-cost images, which have been widely applied to current forestry investigations for precision forestry [11,14,15], especially in complex-terrain areas where traditional forestry field surveys are of low efficiency and high labor intensity [16].

Individual tree detection algorithms for UAV images using image processing or computer vision techniques primarily consist of two types: canopy-boundary segmentation and treetop detection [10,17]. The canopy-boundary segmentation method analyzes the orthophoto after UAV-remote-sensing processing. There are several methods for canopy-boundary segmentation, such as the region growth method [18,19], the watershed segmentation method [20,21], the inverse watershed segmentation method [22], the object-oriented classification method [23], etc. This type of method exploits the differences in the textures and contour features presented in different kinds of objects on remote-sensing images to delineate the canopy boundary. The treetop detection method is based on the principle that the spectral or height value of the treetop at the canopy position of the remote-sensing image is greater than that of the surrounding area. The most commonly used method to detect treetops is the local-maximum (LM) algorithm [24,25], which is both computationally efficient [26] and effective at detecting coniferous treetops [27]. This type of method can obtain both the heights and locations of trees [28].

In the treetop detection method, the traditional local-maximum algorithm is a technique that employs a fixed retrieval window size to extract the local maxima of the luminance or height from remote-sensing images of the forest as the tree location. Early stages of the method detected the spatial locations of individual trees by retrieving the high-reflectivity area in a specific size window on the aerial-remote-sensing image and then marked the pixel point with the highest brightness [24,29]. Currently, the local-maximum method mainly utilizes high spatial-resolution UAV images and LiDAR images to derive a 3D canopy-height model (CHM) to extract more accurate treetops [30,31]. The method can achieve more than 90% accuracy in treetop identification in forest plantations in areas with flat and low canopy densities [32].

However, the CHM-based local-maximum method can still be improved in treetop detection, such as in terms of overlapping crowns, irregular crown shapes, mixed forests with uncertain edges, etc., which complicate the canopy segmentation and can reduce the accuracy of treetop detection [33]. The traditional local-maximum method particularly depends on the search-window-size setting. For example, if the crown sizes of the trees in the recognition area vary significantly, then the algorithm cannot automatically adapt to the crown-size variation in the area, resulting in an inappropriate window size that causes a significant commission error [30,34]. To address this challenge, the local-maximum algorithm with variable windows was developed to enhance the capability of the treetop detection of the LM method in high-density areas and areas with highly varied crown sizes [35,36]. However, the local-maximum algorithm with variable windows does not adapt well to diverse tree-canopy features and generates significant commission and omission errors [10]. It remains unclear whether the LM algorithm can be adapted to the urban environment.

This study aims to propose an improved local-maximum algorithm to detect individual urban trees and retrieve tree heights using UAV images. We developed a framework consisting of three steps: (1) the use of a dual Gaussian filter to reduce the noise in the crown spatial domain and tree-height domain of the canopy-height model (CHM); (2) the optimization of the searching window size of the local-maximum algorithm based on a clustering method to detect as many treetops as possible; and (3) the proposal of the normalized correlation coefficient for treetop detection during the fine-extraction process. The accuracy of our proposed method was compared to the traditional local-maximum

method and validated using a field investigation of sample plots with various canopy closures and a mixture of coniferous, deciduous, and evergreen broadleaved trees.

## 2. Materials and Methods

### 2.1. Study Area

Our study area was located in Shanghai (Figure 1), which has a typical northern subtropical monsoon climate with an annual mean air temperature of 16 °C and precipitation of 1200 mm. The original natural vegetation in Shanghai is characterized by mixed forests composed of subtropical evergreen and deciduous broadleaved trees. However, it has been largely destroyed because of long-term and intense human activity. The current vegetation is dominated by artificial forests and urban green spaces, with considerably high species diversity [37]. To test our improved algorithm, we deliberately selected a part of the area at the plant of the Baosteel Co., Ltd., as our target area for the UAV survey (Figure 1b). The green space of the Baosteel Co., Ltd., is representative of urban green spaces in Shanghai in terms of its diverse plant species and high management activities.

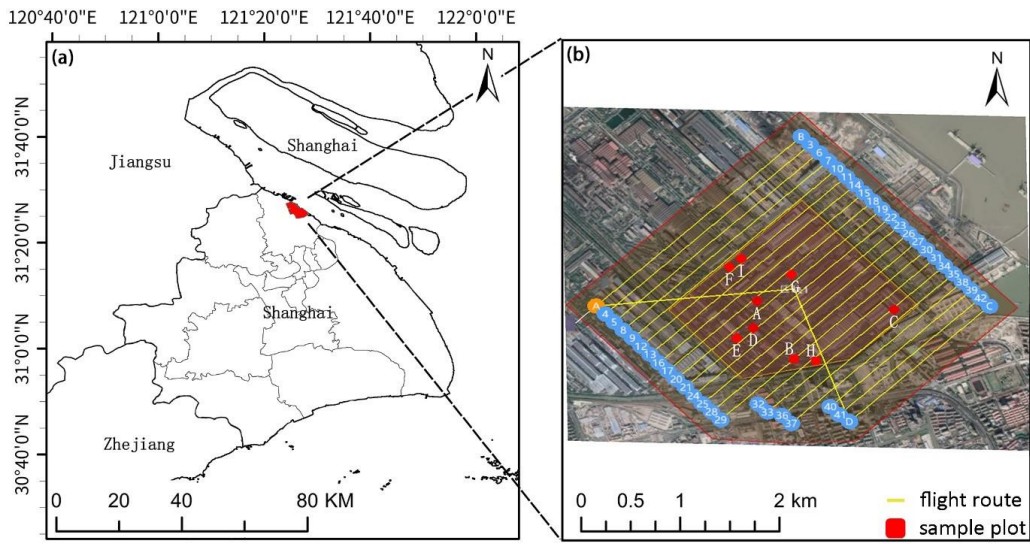

**Figure 1.** (**a**,**b**) Location of the study area and UAV flight route. The A to I is the No. of each field survey sample plot. The blue numbers refer to the No. of the pre-set flight route of the UAV.

### 2.2. Aerial Image Collection and Pre-Processing

The aerial images were collected using a Sony a6000 camera mounted on a FEIMA D2000 four-rotor unmanned aerial vehicle (UAV) made by Shenzhen Feima Robotics Co., Ltd., equipped with Trimble R2 Real-Time Kinematic (RTK) technology. The camera has a 24 MP CMOS sensor with a chip size of 23.5 × 15.6 mm (aps-c) and a lens focal length of 25 mm, which is made by Sony Corp., Wuxi, China. The images were acquired on 31 July 2020 under cloud-free and low-wind-speed weather conditions by the UAV-mounted camera flying along the pre-set flight route (Figure 1b) at a height of 319 m to avoid the higher buildings and production equipment in the plant. The camera took images in RGB mode and set a heading overlapping of 80% and lateral overlapping of 60%, with a high spatial resolution of 0.05 m, which can provide more detailed information about tree species, including the spectral reflectance, texture, color, and crown geometry of the tree canopy. We took a total of 1214 aerial images covering the target area of 2.05 km$^2$. Before the flights, 33 spatially evenly distributed points located at the crosses of roads and/or corners of buildings were identified and set up as the ground-control points (GCPs) using Trimble R2 differential GPS (DGPS) with plane- and elevation-positioning accuracies of 0.02 m and 0.05 m, respectively. Among them, 23 GCPs were used for aerial-photo geo-registration, while the remaining 10 GCPs were used as validation.

The UAV images were first pre-processed, including image geo-referencing and mosaic, color blending, and orthorectification, and were then used to derive the digital orthophoto map (DOM) and canopy-height model (CHM). To generate the CHM, we first produced the digital surface model (DSM), representing the landscape elevation above sea level with objects like trees or buildings, which was generated by the Triangulated Irregular Network (TIN). The TIN was created by three-dimensional point clouds, which were automatically extracted in the embedded DEM module of PHOTOMOD software (version 6.5). Secondly, we produced the digital elevation model (DEM), representing the natural landscape elevation above sea level without objects, by removing all points of buildings and vegetation above the ground surface from the DSM using the embedded filtration tool in PHOTOMOD [38]. During this step, there were still some residual points from the incompletely removed buildings and vegetation, and they had to be manually removed via visual interpretation. Finally, we obtained the DEM, which was used as the datum plane. The rasterized CHM was produced by subtracting the DEM from the DSM [34]. The DOM was orthorectified based on the DEM by using the orthorectification module in PHOTOMOD. It can display the distinguishable boundaries of tree canopies, which can be validated via the visual interpretation of each tree canopy in the sample plots. The spatial resolutions for the DOM and CHM were 0.05 m and 0.1 m, respectively. The accuracy of the CHM data was $\pm 0.126$ m, which was validated from 10 GCPs. All these pre-processing steps were conducted in PHOTOMOD.

### 2.3. Field Investigation

To obtain the validation data for the improved local-maximum algorithm, we conducted a field survey of sample plots that were randomly selected in the green space within the target area (Figure 1b). We performed the field investigation on the plant biodiversity in Baosteel using a sample plot of 20 × 20 m, considering the comparisons with our previous studies, and we selected the previous plot size of 20 × 20 m for our field investigation. First, we overlapped our target area onto the digital layer of the previous 84 sample plots, and we found that 9 of them were located within the target area. Second, we conducted a field survey from 21 September to 3 November 2020. The coordinates of four corner points for each sample plot were primarily recorded using a Garmin eTrex 329x portable GPS with a positioning accuracy of nearly 3 m. Then, the four points were spatially well aligned with the super-high-resolution UAV images to position the precise boundary of each plot in the field. The forests in the 9 selected sample plots contained mixed coniferous and broadleaved stands. The canopy closure (CC) of each plot was estimated via visual interpretation from the UAV digital image using ArcGIS software (Version 10.7) [39], and was briefly categorized into two groups: CCs less than 0.5, including the plots A–D, and CCs from 0.5 to 1.0, including the sample plots E–I. Individuals with diameters at breast height (DBHs) larger than 5 cm and heights of more than 2 m were recorded with their basal positions, heights, DBHs, canopy widths, and scientific taxa and abundances. Each tree species was identified according to the flora of Shanghai [40].

A total of 19 tree species and 554 individual trees were recorded, including 4 conifer species: *Metasequoia glyptostroboides*, *Cedrus deodara*, *Juniperus chinensis* 'Kaizuka', and *Chamaecyparis pisifera* 'Squarrosa', and 15 broadleaved tree species. The diameters at breast height (DBHs), tree heights (THs), and crown widths (CWs) for all tree species in the nine sample plots were in the ranges of 5.7–57.3 cm, 4–30 m, and 2–10.5 m, respectively. Each of these parameters for the individual trees was recorded for the validation of the estimation.

### 2.4. Improvement to Local-Maximum Algorithm

We utilized the local-maximum (LM) algorithm to detect treetops to count the number of trees in each plot. However, the traditional LM suffers from the fixed searching-window-size of the tree crown, which cannot deal with the canopy complexity, such as crown overlapping, tree-crown irregularity, mixed canopies with evergreen and deciduous trees, etc., especially the true- and pseudo-treetop questions. To solve these problems, we pro-

posed an improved algorithm framework: the local-maximum correlation (LMC) algorithm, which is suitable and robust for detecting the treetops of urban forests, especially mixed evergreen and deciduous broadleaved forests. Our improvement to the LM mainly included the following aspects: First, the canopy-height model was optimized via dual Gaussian filtration to reduce the noises from the original CHM input data. Second the searching window was generated via a clustering analysis of the tree height and crown size in order to find as many tree vertices as possible. Third, the crown-profile model was used to make the largest and smallest crown templates as the reference to identify the true and pseudo treetops using the normalized correlation coefficient (NCC). The overall workflow is shown in Figure 2. A detailed description of each improvement can be found in the following sections.

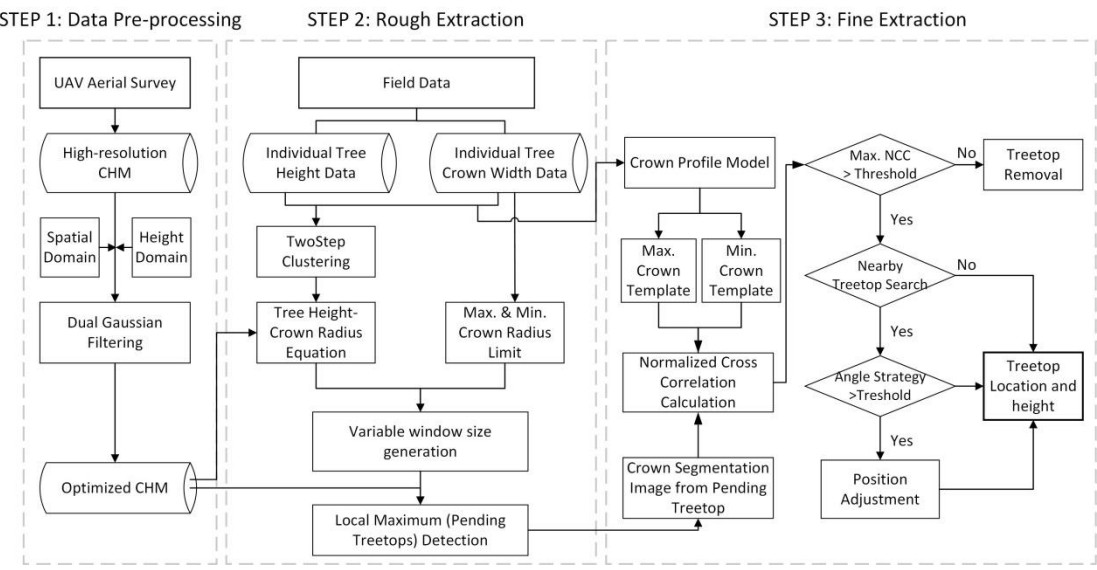

**Figure 2.** The flowchart of treetop detection using the improved LM algorithm.

The algorithm framework was developed on the R Studio platform (Version 1.22.5033) in the R (Version 4.0.2) environment. The local-maximum detection in step 2 was improved from the "vwf" function in the "ForestTools" package [41]. The algorithm did not use other libraries except for the "raster" library for geographic data processing [42]. We developed the programming for each algorithm in our framework, such as dual Gaussian filtering, the canopy maximum and minimum tree-crown templates, normalized correlation coefficient calculation, and the angle-filtering strategy.

### 2.4.1. CHM Optimization Using Dual Gaussian Filtering

When attempting to find the local maxima directly from the original canopy-height model (CHM), individual treetop detection may present an issue known as 'pseudo-treetop.' This term refers to a situation in which more than one local-maximum value may be detected within a single tree crown. This is often the case with broadleaved forest species, which frequently have expansive crowns composed of several sturdy and lateral branches. Consequently, this complicates the detection process and necessitates the implementation of data pre-processing measures. The frequently used pre-processing method is CHM filtration to smooth the noises on the surface of the canopy and to reduce the fluctuation in its height value [43]. However, these filtering methods, such as the mean-value filter and Gaussian filter, can also simultaneously smooth out smaller trees with lower heights and coniferous trees with crowns that are highly intersected with the surrounding tree crowns.

Therefore, the filtering method could lead to omission and commission errors in treetop detection. Dual Gaussian filtering was proposed to address this issue [33]:

$$I^{filtered}(X) = N^{-1}(X) \int_{-\infty}^{\infty} \int_{-\infty}^{\infty} I(\varepsilon)(H(I(\varepsilon), I(X)) + G(\varepsilon, X))d\varepsilon \tag{1}$$

$$N(X) = \int_{-\infty}^{\infty} \int_{-\infty}^{\infty} (H(I(\varepsilon), I(X)) + D(\varepsilon, X))d\varepsilon \tag{2}$$

$$H(I(\varepsilon), I(X)) = e^{-\frac{\|I(\varepsilon) - I(X)\|}{2\sigma_h^2}} \tag{3}$$

$$D(\varepsilon, X) = e^{-\frac{\|\varepsilon - X\|}{2\sigma_d^2}} \tag{4}$$

where $I(X)$ is the original input image to be filtered; $I^{filtered}(X)$ is the processed CHM image; $X$ is the current calculated pixel point; $\varepsilon$ is the surrounding neighborhood pixel points around $X$; $N(X)$ is the normalization factor. $H(I(\varepsilon), I(X))$ and $D(\varepsilon, X)$ are Gaussian functions based on the height-value domain and spatial domain, respectively, where $\sigma_h$ and $\sigma_d$ are the smoothing parameters in these two functions, respectively.

Three parameters, the window size, $\sigma_d$, and $\sigma_h$, need to be adjusted during this filtering. The window size was defined using the average crown width of 4 m, which was calculated from the field inventory data of all individual tree-crown sizes. This ensured that large tree crowns would be smoothed while tree edges would be kept. The values of $\sigma_d$ and $\sigma_h$ should be correlated with the CHM height values in order to achieve good data pre-processing demands ($\sigma_d = a2 \times \sigma_h = a1 * I(X)$). The sizes of the $\sigma_d$ value and $\sigma_h$ value determine the effect of the dual Gaussian filter in crown spatial smoothing and edge protection, respectively. The smaller the values, the better the Gaussian-filtering effects on smoothing and edge protection for the neighboring pixels compared with distant pixels. Broadleaved trees with larger crowns require a larger Gaussian-filter range, while smaller trees with smaller crowns should have a smaller Gaussian-filter range. It was found that a1 = 0.055 and a2 = 2 work better. The filtering effect is shown in Figure 3.

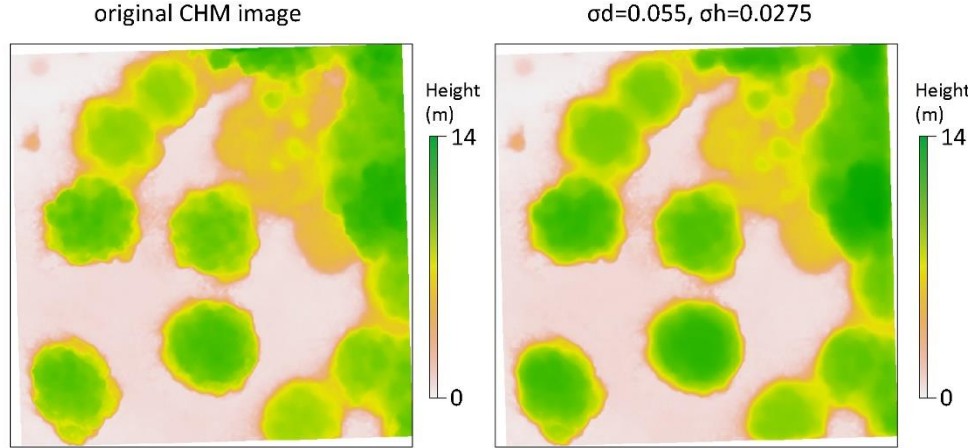

**Figure 3.** Smoothing effect on the noise of the CHM via dual Gaussian filter. The heights and edges of tree crowns look clearer and smoother after filtering (**right image**) compared with the original-input CHM image (**left image**).

### 2.4.2. Selection of Suitable Window Size for Improved LM Algorithm

After the dual Gaussian filtering to the CHM, we used the local-maximum detection algorithm to identify the treetops from the CHM. The local-maximum algorithm works on the assumption that the highest value in a local area or a spatial neighborhood on the CHM represents the tip of a tree crown [26,44,45]. Therefore, it is crucial to select

a suitable window size for the local-maximum filter. Too large or too small a window size will result in omission or commission errors, respectively [24,26]. To better detect treetops, the variable window size has been widely used, and the window can be square- or circle-shaped [24–26,46]. In this study, we selected the square-shaped variable window size due to the fact that, firstly, most of the forest stands in our sample plots were mixed forests with multiple species and varying crown sizes, and the single or fixed window size is not enough to capture the complex morphology of the forest canopy [47]; secondly, filtering for the local maximum with square-shaped windows provides a better model fitting for hardwood species with various crown shapes [26]. The variable window of the local maximum should be adjusted to an appropriate size that corresponds to the spatial neighborhood on the CHM image. Generally, the generation of the appropriate window size for treetop searching is based on the assumption that a relationship exists between the heights of the trees and their crown sizes [26].

Unlike natural vegetation, street trees and forest stands in subtropical cities like Shanghai are usually mixed with coniferous, evergreen, and deciduous broadleaved trees to increase diversity and enhance landscape aesthetics, which challenges the variable windows generated for the local-maximum filter [47]. This study utilized the tree-height and crown-size data from the field investigation to generate the appropriate window size through the following steps: Firstly, the tree-height and tree-crown data were classified using two-step cluster analysis. Two-step clustering is a machine learning technique that can automatically divide a given dataset into multiple specific subgroups in which data in the same subgroup have similar attributes and features [48]. There were a total of 554 individual trees, and they were finally clustered into two groups using their height and crown-size data obtained from the field investigation. Group I had 329 trees with heights less than or equal to 12 m; Group II had 225 trees with heights of more than 12 m (see Table 1).

**Table 1.** General statistics of the height–crown radius cluster groups of trees in all sample plots.

| Group | Number | Indicator | Minimum (m) | Maximum (m) | Mean (m) | S.D. | Group |
|---|---|---|---|---|---|---|---|
| ≤12 m | 329 | Tree height | 3.00 | 12.00 | 7.95 | 2.35 | 3.00 |
| | | Canopy radius | 0.25 | 5.50 | 1.96 | 0.84 | 0.25 |
| >12 m | 225 | Tree height | 12.00 | 38.00 | 20.12 | 5.42 | 12.00 |
| | | Canopy radius | 0.88 | 5.75 | 2.57 | 0.95 | 0.88 |

Secondly, the relationship between the tree heights and crown sizes for each clustered group was established using linear or nonlinear fitting, and the optimal fitting model was the quintic polynomial model when evaluated using the Akaike information criterion (AIC). The models were used to predict the crown width (y) using the height value (x) from the CHM (see Table 2). The predicted crown sizes were then used to generate the filtering window size.

**Table 2.** Polynomial fitting of crown diameter and tree height for the two groups of sample plots.

| Group | Fitting Equation | $R^2$ | AIC |
|---|---|---|---|
| ≤12 m | $y = -2.86598 \times 10^{-4}x^5 + 8.27 \times 10^{-3}x^4 - 0.08274 \times x^3 + 0.32921 \times x^2 - 0.29568 \times x + 0.05897$ | 0.8896 | $\overline{72.15}$ |
| >12 m | $y = 1.86478 \times 10^{-6}x^5 - 2.17364 \times 10^{-4}x^4 + 9.4 \times 10^{-3} \times x^3 - 0.18446 \times x^2 + 1.62422 \times x - 3.87244$ | 0.5207 | $\overline{53.35}$ |

Thirdly, to further improve the variable window of the local-maximum algorithm during the rough-extraction stage in order to find the exact pending treetops, and as many

as possible, the size of the generated variable window ($S_{vw}$) above was further limited using Equation (5):

$$S_{vw} = \begin{cases} minRadius & vw < minRadius \\ fitting\ equation & minRadius < vw < meanDiameter \\ meanDiameter & vw > meanDiameter \end{cases} \quad (5)$$

For the maximum variable window size, the mean value of the crown width was 3.2 m, which was calculated from the field survey data and was chosen as the limit. For the minimum variable window, the minimum crown width was selected. In addition, elevation values below 1.5 m were ignored for the local-maximum detection of the CHM to exclude shrubs, ground cover, and young trees in the forest gaps in urban areas.

2.4.3. Fine Extraction

A larger number of pending treetops are found in the rough-extraction stage, and these local maxima are determined one by one in the fine-extraction stage to see whether they should be kept, removed, or merged. Due to the considerable crown variation in mixed coniferous and broadleaved forests in subtropical urban areas, we used the normalized correlation coefficient method to check the similarity between the crown size of each of the pending treetops and the crown width in the crown template with the ideal maximum/minimum values of that tree species. Based on the similarity, we decided whether to keep the pending treetops or not. Later, we checked the morphology of other pending treetops in the neighborhood and used the angle-filtering strategy to finalize the treetop selection.

The crown-profile models were fitted to the largest and smallest canopy species in the study area (in this study, *Cinnamomum camphora* and *Metasequoia glyptostroboides*, respectively). These 3D models' highest points corresponded to the pending treetop location on the CHM, and the height value and window size of the pending treetop were used to generate canopy templates of the ideal maximum- and minimum-canopy-radius tree species for that point (Figure 4). Then, the normalized correlation coefficient (NCC) was calculated by comparing these two types of templates with the canopy-plane image of the pending treetop [49]:

$$\gamma(x,y) = \frac{\sum_s \sum_t [w(s,t) - \bar{w}] \sum_s \sum_t [f(x+s, y+t) - \bar{f}_{xy}]}{\left\{ \sum_s \sum_t [w(s,t) - \bar{w}]^2 \sum_s \sum_t [f(x+s, y+t) - \bar{f}_{xy}]^2 \right\}^{\frac{1}{2}}} \quad (6)$$

where $\gamma(x,y) \in [-1,1]$ is the NCC matrix of the CHM image; $w(s,t)$ is the plane matrix made for crown-image mapping to the crown-profile model; $f(x+s, y+t)$ is the plane matrix of the labeled crown image extracted from the rough-extraction stage for the pending treetops; and $\bar{f}_{xy}$ is the mean of the entire crown image of the pending treetops.

The two NCCs calculated for each treetop represent the extent to which the morphology of that treetop fits the ideal crown morphology within the detection window. The higher of the two values is chosen as the correlation result, and if the treetop NCC is above the threshold, then the point is confirmed as the actual treetop. If the treetop NCC is below the threshold, then the point will be removed.

We searched for other treetops that existed within the window of the filtered treetops, after which the detected treetops within the window were calculated two by two. We created a line connecting the two points, found the lowest point presented on the surface of the canopy on that path, and calculated the angle between the crowns of the pending treetops. Based on knowledge of crown morphology, two pending treetops may be local maxima on the same broadleaf tree when the angle is above the threshold. The pending treetop with the lower NCC is removed, and the spatial position of the treetop with the higher correlation coefficient will be adjusted. When the angle is below the threshold, the two treetops can be considered as belonging to two trees.

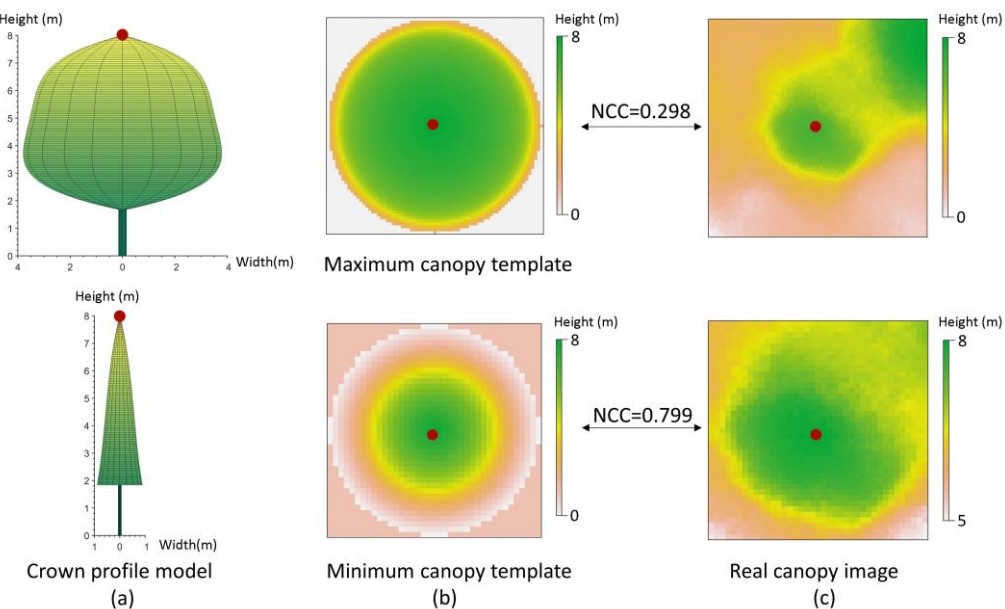

**Figure 4.** Schematic diagram illustrating the use of a crown-profile model to generate a crown-plane template to calculate normalized correlation coefficients (NCCs). (**a**) A 3D crown-profile model of the maximum and minimum canopy tree species in the study area. (**b**) Crown-plane template of a specific treetop calculated from (**a**), with the crown-radius and height-value parameters obtained via the LMC method in the rough-extraction stage. (**c**) NCC calculation between the actual tree-crown image and crown-plane template to quantify the similarity.

### 2.5. Accuracy Assessment

According to the accuracy assessment methods utilized by previous studies [10,50], the accuracies of the treetops detected by the LMC were validated using commission errors (CE), omission errors (OE), the user accuracy (UA), the producer accuracy (PA), the detection ratio (DET), and the overall accuracy (OA) at the sample-plot scale. A treetop detected via the LMC algorithm is considered correct if the distance between the detected treetop and the actual treetop is less than 1/2 of the maximum canopy width. The accuracy indices were calculated using the following equations:

$$\mathrm{CE} = \frac{N_c}{N_v} \times 100\% \tag{7}$$

$$\mathrm{OE} = \frac{N_o}{N_v} \times 100\% \tag{8}$$

$$\mathrm{UA} = (1 - \mathrm{CE}) = \left(1 - \frac{N_c}{N_v}\right) \times 100\% \tag{9}$$

$$\mathrm{PA} = (1 - \mathrm{OE}) = \left(1 - \frac{N_o}{N_v}\right) \times 100\% \tag{10}$$

$$\mathrm{DET} = \left(\frac{N_d - N_c}{N_d}\right) \times 100\% \tag{11}$$

$$\mathrm{OA} = 1 - \frac{|N_d - N_v|}{N_v} \times 100\% \tag{12}$$

where $N_d$ denotes the number of urban trees identified via the LMC method; $N_v$ denotes the actual number of trees present on the site; $N_c$ is the number of trees incorrectly marked via the LMC method; and $N_o$ is the number of trees not detected via the LMC method. The CE represents the probability that the real data are misclassified, and the UA represents the

percentage of correct detections within the ground-truth data after classification. The OE represents the probability of being missed in the ground-truth data classification, and PA represents the probability of being detected in the ground-truth data after classification. The DET represents the probability that the detection is correctly validated against the ground-truth data. The OA is the ratio of correct detection to the ground-truth data.

For comparison, we also detected treetops using the original local-maximum algorithm in the R package "ForestTools" [41]. A one-way ANOVA was used to check the differences between the sample plots with different canopy closures. The accuracies of the estimated tree heights were assessed via the field investigation data using linear regression and its R-squared and root-mean-squared error (RMSE) [51].

## 3. Results

### 3.1. Treetop Detection and Accuracy Assessment

The accuracy validation data for the LMC algorithm are shown in Table 3. For all the sample plots, the average overall accuracy (OA) reached 79.3%, the average detection rate (DET) reached 85.1%, the average user accuracy (UA) was 82.5%, the average producer accuracy (PA) was 77.8%, and the average commission error (CE) and omission error (OE) were 17.5% and 22.2%, respectively. The accuracy depends on the canopy closure, where less canopy closure generally leads to higher accuracy. The accuracy indices for samples with less canopy closure were relatively higher and their standard deviations were considerably smaller than those with larger canopy closures, which demonstrates that the robustness of our algorithm may degrade with an increase in the canopy closure.

**Table 3.** Accuracy verification of treetop detection in the nine sample plots.

| Sample Plot | Canopy Closure | Actual Tree Number | NDT | CE (%) | OE (%) | UA (%) | PA (%) | DET (%) | OA (%) |
|---|---|---|---|---|---|---|---|---|---|
| A | 0.5 | 84 | 93 | 13.98 | 23.66 | 86.02 | 76.34 | 84.52 | 90.32 |
| B | 0.5 | 28 | 28 | 17.86 | 17.86 | 82.14 | 82.14 | 82.14 | 100 |
| C | 0.5 | 34 | 37 | 13.51 | 13.51 | 86.49 | 86.49 | 85.29 | 91.89 |
| D | 0.5 | 153 | 168 | 5.36 | 13.69 | 94.64 | 86.31 | 94.12 | 91.07 |
| Mean (±SD) | | | | 12.68 (±2.55) | 17.18 (±7.12) | 87.32 (±7.55) | 82.82 (±2.12) | 86.52 (±6.54) | 93.32 (±3.90) |
| E | 1 | 49 | 42 | 26.19 | 9.52 | 73.81 | 90.48 | 77.55 | 83.33 |
| F | 1 | 89 | 121 | 4.96 | 31.4 | 95.04 | 68.6 | 93.26 | 73.55 |
| G | 1 | 65 | 43 | 74.42 | 23.26 | 25.58 | 76.74 | 50.77 | 48.84 |
| H | 1 | 85 | 122 | 0.82 | 31.15 | 99.18 | 68.85 | 98.82 | 69.67 |
| I | 1 | 111 | 171 | 0.58 | 35.67 | 99.42 | 64.33 | 99.1 | 64.91 |
| Mean (±SD) | | | | 21.39 (±31.47) | 26.2 (±10.34) | 78.61 (±31.47) | 73.8 (±10.34) | 83.9 (±20.49) | 68.06 (±12.70) |
| OM | | | | 17.52 | 22.19 | 82.48 | 77.81 | 85.06 | 79.29 |

Note: NDT: the number of treetops detected using the LMC method. OM: mean accuracy of the algorithm over all sample plots.

Figure 5 shows the treetop detections using the local-maximum (LM) and LMC algorithms for the nine sample plots with different canopy closures and tree species distribution patterns. In the sample plots Figure 5A–D, with canopy closures of about 0.5, the LMC algorithm captured fewer pending treetops than the LM algorithm and resulted in more accurate treetop detections and a lower CE. The LMC algorithm was robust with lower standard deviations, and the average OA was 93.3% (±3.9) (see Table 3). In contrast, for the sample plots Figure 5E–I with canopy closures close to 1, the accuracies of the treetop detections for both the LM and LMC algorithms were greatly reduced. The average OA was 68.1% (±12.7). The LMC algorithm could detect fewer broken canopy points located on the edges of the sample plots than the original LM algorithm. In particular, within the sample plot labeled 'G,' a significant number of European oleander (*Nerium oleander*)

trees were present. These trees are characterized by their tendency to grow numerous branches, leading to a relatively flat canopy area replete with multiple robust branch tops. This characteristic of oleanders had a noticeable impact on the accuracy indices. Specifically, the commission-error (CE) value and omission-error (OE) value were found to be 74.4% and 23.3%, respectively, indicating a notable decrease in accuracy when compared to plots with less complex canopies. The average CE was 21.4% ($\pm$31.5) and the average OE was 26.2% ($\pm$10.3) for the sample plots with higher canopy closures.

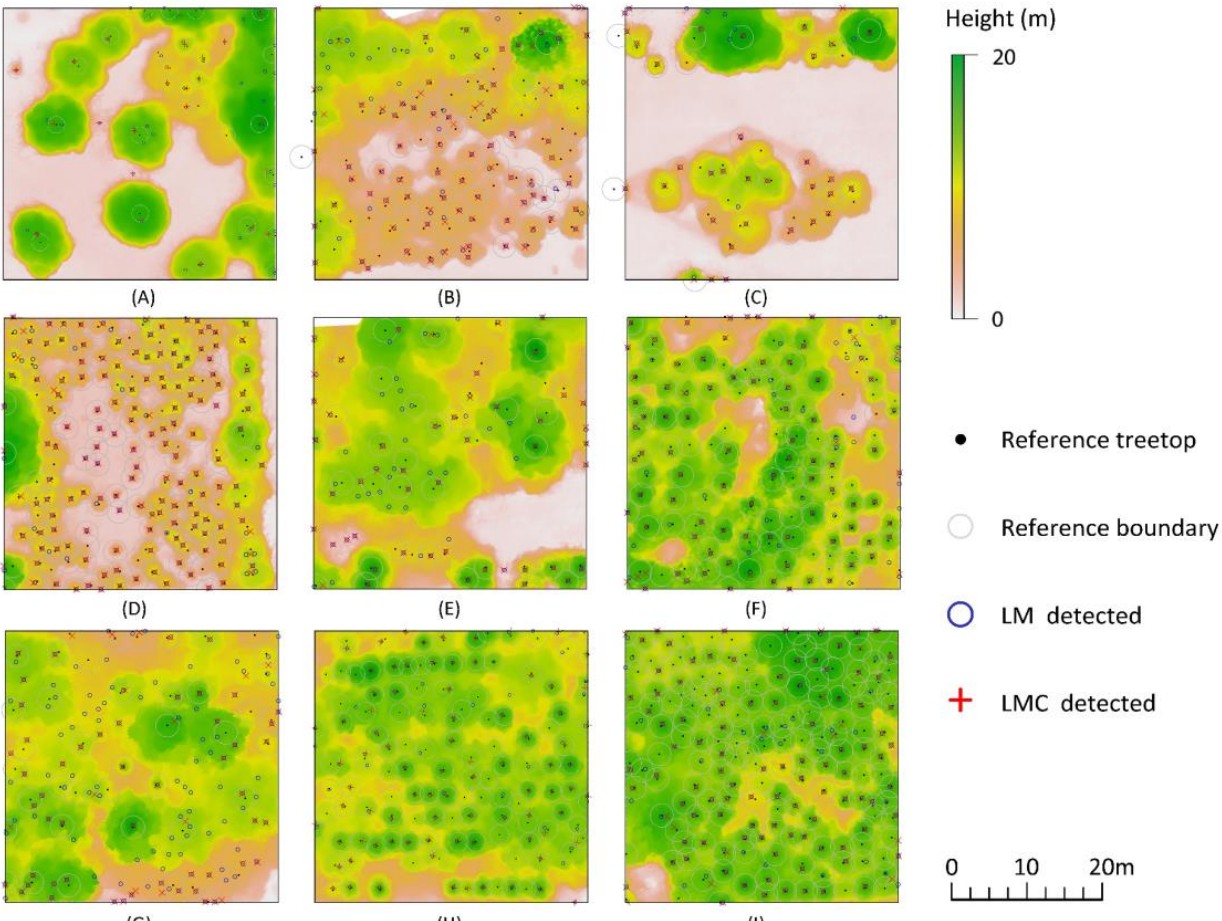

**Figure 5.** Treetops detected in each sample plot. The blue circle represents the treetop position detected using the original LM algorithm. The cross represents the treetop position detected using the LMC method. Black points mark the reference treetops positioned via field investigation; grey circles refer to the extent of tolerance for the positional deviations between the detected treetop and the actual referenced treetop when the treetop point is not overlapped with the point of the trunk basal due the fact that the trunk may grow at an angle, or the highest point of the canopy may lie on a strong branch of the canopy. The size of the sample plots was 40 $\times$ 40 m. The stands were mixed conifer–broadleaf forests in each plot. The canopy density was nearly 0.5 in the plots (**A–D**), and approximately 1 in the plots (**E–I**). Plot G had a lot of Nerium oleander trees with flattened canopies composed of several strong branches.

### 3.2. Tree-Height Estimation and Accuracy Evaluation

The tree-height estimates using the LMC algorithm had high accuracy compared with the field inventory. The R$^2$ of the linear fitting of the estimated tree heights for the sample plots with lower canopy densities of 0.5 and 1 reached 0.97 and 0.96, respectively (Figure 6). Compared with the tree-height investigations in the sample plots with canopy densities (CD) of 0.5, the estimations via the LMC algorithm were overestimated with the tree-height range from 5.0 to 10.0 m (Figure 6a), and the average RMSE of the tree-height estimation

was 0.54 m. For the sample plots with CDs close to 1, the overestimation mainly occurred in tree heights between 12 and 20 m, the lower estimation presented in the tree-height range of 5–12 m (Figure 6b), and the average RMSE of the tree-height estimation was 0.67 m.

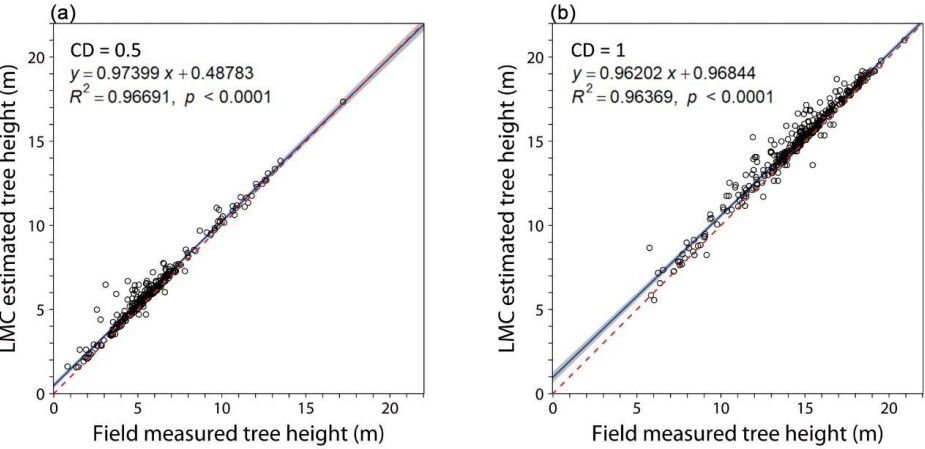

**Figure 6.** The scatterplot of tree-height estimation using the LMC algorithm vs. the field-measured tree height. The graph (**a**) shows the results of the linear fit of tree heights in sample plots Figure 5A–D with canopy densities (CDs) of 0.5. The graph (**b**) shows the results of the linear fit of tree heights in sample plots Figure 5E–I with CDs close to 1.

## 4. Discussion

### 4.1. The Influences of Parameter Settings in the LMC Algorithm

In subtropical urban areas, it is common to find broadleaved species intermixed with coniferous species in the urban landscape. Given this mixed-species environment, it becomes essential to consider the impacts of the algorithm parameter settings on the tree species. Specifically, attention should be given to trees with varying crown radii and heights during each step of the algorithm's execution. Our results showed that, in the data pre-processing step, the optimal parameter setting for the dual Gaussian filter can result in a desirable smoothing of the height values of the crowns with different spatial distributions of heights. This goal is well achieved for the canopy areas in the CHM image where a lot of broadleaved tree species stand with high treetops of their crowns; therefore, a smaller $\sigma_d$ parameter setting is needed to increase the spatial smoothing effect, which can well suppress the local maxima generated by a number of strong branches within the crown and, hence, reduce the commission errors. In addition, a smaller $\sigma_d$ also can reduce the variance in the height values, which also helps to increase the local angle on the canopy and then enhance the filter selection in the next step during the angle detection. For canopy areas with lower height values where a lot of crown edges of larger trees and small trees are predominantly located, a smaller $\sigma_h$ value can facilitate the suppressing of the smoothing effect on the crown edges and ensure the angle detection to maintain the external crown forms of these small trees. For instance, in our study, for the CHM image, when a1 = 0.055 and a2 = 2, and $\sigma_d = 0.055$ and $\sigma_h = 0.0275$, a good combined smoothing effect on the edges and heights could be achieved.

The aim of our improved LMC method is to obtain as many accurate treetops as possible during the rough-extraction stage. Our variable window size could well identify more treetops on the CHM image by clustering the target tree species into groups and simulating different patterns of crown morphologies of coniferous and broadleaved trees in subtropical urban areas. The window size was finally adaptively estimated using a five-order polynomial fitting [47]. To ensure that as many accurate pending treetops as possible could be obtained, a piecewise function using a minimum and maximum limit was also useful to generate the variable window size, which benefited the further crown-width selection during fine extraction.

In the fine-extraction stage, the CHM data are used to determine whether the tree-top candidates obtained in the previous stages should be reserved, merged, or removed. Therefore, it is important to form a more accurate crown template to calculate the exact correlation values for each pending treetop. The spatial location of the treetop candidates obtained via the local maximum and the window size in which the point was extracted were used to generate the canopy-plane template. To distinguish the pattern of the crown sizes of different urban tree species, a crown-profile template was created for the tree species with the largest crown sizes (i.e., the *Cinnamomum camphora*) and for the tree species with the smallest crown size (i.e., the species *Metasequoia glyptostroboides*) in the sample plots, and then the NCC range could be calculated between the crown form of the pending treetops and the largest and smallest crown templates. As the crown-profile template produced was a planar form of the entire crown of that tree, direct use of the searching window size generated in the rough-extraction stage to extract the true crown may have resulted in underestimation. We tested this by applying 2.25 times the searching window size for the broadleaved-tree-crown template and 1.2 times the searching window size for the conifer-tree-crown template, and with an NCC threshold of 0.4, which produced more robust treetop selection results. It is possible that some of the selected pending treetops may have still been one of the local maxima within the same large tree crown; therefore, they needed to be subsequently merged and deleted using an angle-filtering strategy. As the horizontal distance between two treetops increases, the angle between the two points and the connected minimum point increases, which indicates that the angle-filtering mechanism will not work when the two points are far away from each other. Therefore, we excluded treetops with horizontal distances between the two points above the average crown width of 3.2 m and set the angle over 120° as the angle-filtering threshold, thereby reducing the probability of error in merging treetops while ensuring the feasibility of the angle-filtering strategy.

### *4.2. Error Analysis*

There are several factors that could have affected the accuracy of the treetop detection using the LMC algorithm. Firstly, some of the trees located at the edges of the sample plots resulted in broken and incomplete crowns when cutting the CHM image into the recognition square. These incomplete crowns were detected as potential treetops and could thus enhance the commission error. Secondly, certain tree species, such as the European oleander (*Nerium oleander*) found in sample plot G, exhibit a cespitose branching pattern, while others, like the camphor tree (*Cinnamomum camphora*) in sample plot E, have large crowns with several robust branches. The tops of these branches have only slight height differences compared to the actual treetops. This subtle difference poses challenges in distinguishing between them, which can lead to commission errors in the treetop identification process. Thirdly, errors can occur in the creation of the canopy-height model (CHM). The CHM is produced by deducting the elevation of roads and lawns in the digital elevation model (DEM) from the canopy height derived from the point-cloud data obtained from unmanned-aerial-vehicle (UAV) images. This process may inadvertently reduce the recorded heights of some young trees that are slightly taller than 1.5 m to less than 1.5 m. As a result, these trees could be completely omitted in the CHM. This issue is evident in sample plots F, H, and I, which were utilized as final input data. Consequently, this leads to what is known as an omission error. Lastly, due to the inherent defect of the optical sensor of the UAV onboard, some of the smaller and lower trees under the large trees could not be monitored and therefore could not be detected, resulting in omission error.

In addition, previous studies have demonstrated that the canopy closure and slope of the study area could influence the accuracy of treetop detection [10,16]. The overall accuracy of treetop detection is in the range of from 78% to 96.4%, corresponding to canopy closures of high and low, respectively [10,52], which is similar to our study, with an overall accuracy range from 68.1% for a high canopy closure of nearly 1.0 to 93.3% for a canopy closure less than 0.5, but much higher than that of the original LM algorithm with an

overall accuracy of 61.0%. Our study showed that the canopy closure also influenced the accuracy. There was a strong correlation between the canopy closure and the overall accuracy (OA) when examined using the one-way ANOVA method. Table 4 shows that there is a significant difference between the production accuracy (PA) and OA for the sample plots with different canopy closures ($p < 0.05$). This indicates that, for the LMC algorithm, the canopy closures of the sample plots had a large impact on the omission error, especially the OAs of the sample plots.

**Table 4.** Effects of different canopy densities on accuracy of sample-plot verification data.

| Statistical Measure | NDT | Nv | UA | PA | DET | OA |
|---|---|---|---|---|---|---|
| F | 0.032 | 0.208 | 0.292 | 2.552 | 0.061 | 14.055 |
| *p*-value | 0.863 | 0.662 | 0.605 | 0.154 | 0.813 | 0.007 |

F is the statistical measure of the F test. The homogeneity test for each dependent variable had *p* values > 0.05, which satisfied the homogeneity of variance.

In comparison to other studies that utilize UAV-derived CHMs for individual treetop detection, our improved local-maximum correlation (LMC) algorithm demonstrates improved accuracy. For instance, when applying the LM algorithm to primeval temperate forests in Mazandaran Province, Iran, Ahmadi et al. achieved a tree detection accuracy of only 0.60 [53], which is significantly lower than our accuracy of 0.793. Similarly, research conducted in Valongo, Porto, Portugal, employing UAV Structure-from-Motion (SfM) technology to generate point clouds, yielded an 80% detection accuracy [54], which is similar to our accuracy. But our accuracy is lower than that of another study conducted in the Roda River catchment, Jena, Germany, in which the detection rate and commission error reached 93.2% and 10.7%, respectively [55], and of that of the study conducted in a private forest at Cache Creek located in eastern Jackson city in Wyoming, USA, in which the accuracy was more than 85% [28]. The tree detection accuracy is largely dependent on the complex canopy structure and tree species diversity. The mean overall accuracy of our LMC algorithm reached 79.3% and could be considerably acceptable. Therefore, our LMC algorithm is capable of accurately detecting individual trees, even in situations in which the tree crowns are heavily overlapped, or when forest stands exhibit dense canopy cover. It would be valuable to conduct additional studies to assess the applicability of this improved LMC algorithm in different places in diverse locations and cities.

## 5. Conclusions

This research enhances the traditional local-maximum algorithm with several optimization strategies aimed at improving the analysis of canopy-height models derived from UAV imagery for urban forests in subtropical regions, such as Shanghai. During data pre-processing, we employed dual Gaussian filtering to refine the canopy-height model. For the rough-extraction phase, we utilized a clustering method to optimize the generation of the variable window. In the final fine-extraction stage, the local normalized correlation coefficient was used to construct a crown-plane model, aiding in the detection of treetops and tree heights.

The improved local-maximum algorithm demonstrated notable success in treetop detection, with an average user accuracy of 87.3% for lower-density urban forests and 78.6% for high-density urban forests. In addition, the algorithm's tree-height estimation closely aligned with the ground-truth tree-height measurements, evidenced by an $R^2$ value exceeding 0.96 and an average root-mean-square error (RMSE) of 0.61 m across urban forests with varying canopy closures.

The enhanced algorithm outperforms the original local-maximum approach in treetop detection for subtropical urban areas. It shows particular promise for the mapping of subtropical mixed coniferous–broadleaf urban forests, reinforcing its utility in improving urban-forest-mapping accuracy.

**Author Contributions:** Conceptualization, J.L. and C.W.; methodology, H.W. and Y.C.; software, H.W. and C.M.; validation, H.W. and M.Z.; formal analysis, H.W. and M.Z.; investigation, M.Z., C.W., L.O., Y.L. and Y.S.; resources, J.L. and C.M.; data curation, H.W. and M.Z.; writing—original draft preparation, H.W., C.W. and J.L.; writing—review and editing, J.L., C.W. and T.Q.; visualization, H.W.; supervision, J.L. and C.W.; project administration, J.L. and C.W.; funding acquisition, J.L., C.W. and Y.T. All authors have read and agreed to the published version of the manuscript.

**Funding:** This research was supported partially by the National Key R&D Program of China (Grant No. 2022YFF1301105), the National Natural Science Foundation of China (No. 31870453) to J.L. and the National Natural Science Foundation of China (No. 32001162) and China Postdoctoral Science Foundation (No. 2021M702131) to C.W., and the project from Baosteel Co., Ltd. to Y.T. and J.L.

**Data Availability Statement:** The data presented in this study are available on request from the author.

**Acknowledgments:** We are grateful to Jingyao IT Co., Ltd. and Baosteel Co., Ltd. for their assistance with the UAV survey and sample-plot field investigation, respectively.

**Conflicts of Interest:** The authors declare no conflict of interest.

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
