# Peer review of "Urban Treetop Detection and Tree-Height Estimation from Unmanned-Aerial-Vehicle Images"

_remotesensing, doi:10.3390/rs15153779_

Round 1

Reviewer 1 Report

The paper components are well presented, and the purpose of the research meet the objectives of the journal. For these reasons, I believe that the manuscript can be published within Remote Sensing journal after minor revision.

Some minor suggestions and comments are as follows.

1.     L64-65: “object-oriented classification method” is not the method for canopy boundary segmentation.

2.     L143-144: What is the elevation accuracy of the CHM data?

3.     L155: What kind of GPS? What is the positioning accuracy?

Minor English revision is needed. For example, L83: “still can still”?

Reviewer 2 Report

This study discussed treetop detection algorithm based on CHM collected by UAV, targeting on the urban forest in Shanghai. This study provided a new concept for refining present LM treetop detection algorithm by integrating the techniques of a dual gaussian filtering, window size optimization and fine extraction based on 3d crown profile modelling. The manuscript is well written.

I added some minor comments as below.

Line 83

Duplicated “still”

still can still be

Line 95

It remains unclear is LM

It remains unclear that LM

Line 108

Our study area locates in Shanghai

Our study area is located in Shanghai

Line 121-122

Please add some information on UAV.

Line 233

I am not sure that subtitle “2.4.2. Improvement to local maximum algorithm” is appropriate or not. It is same with subtitle “2.4. Improvement to the local maximum algorithm”.

In this section, selection of window shape and size is discussed.

Line 305

Crown plan template

Crown plane template?

Line 374-375

Correct Estimation(CE) value

Over Estimation (OE) value

Area these correct? This is Confusing.

In my opinion, CE is commission errors and OE is omission errors as defined in line 325-326.

Figure 4.

I would like to confirm that the height range of (c) of lower figure is 5 to 8 not 0 to 8.

English is acceptable. Please check some minor points that I mentioned.

Reviewer 3 Report

Please find the pdf in the attachment
